# Learning Dynamical Systems from Noisy Data with Inverse-Explicit Integrators

## Abstract

We introduce the mean inverse integrator (MII), a novel approach to increase the accuracy when training neural networks to approximate vector fields of dynamical systems from noisy data. This method can be used to average multiple trajectories obtained by numerical integrators such as Runge–Kutta methods. We show that the class of mono-implicit Runge–Kutta methods (MIRK) has particular advantages when used in connection with MII. When training vector field approximations, explicit expressions for the loss functions are obtained when inserting the training data in the MIRK formulae, unlocking symmetric and high order integrators that would otherwise be implicit for initial value problems. The combined approach of applying MIRK within MII yields a significantly lower error compared to the plain use of the numerical integrator without averaging the trajectories. This is demonstrated with experiments using data from several (chaotic) Hamiltonian systems. Additionally, we perform a sensitivity analysis of the loss functions under normally distributed perturbations, supporting the favourable performance of MII.

## 1 Introduction

Recently, many deep learning methodologies have been introduced to increase the efficiency and quality of scientific computations [1, 2, 3, 4]. In physics-informed machine learning, deep neural networks are purposely built so to enforce physical laws. As an example, Hamiltonian neural networks (HNNs) [5] aim at learning the Hamiltonian function from temporal observations. The Hamiltonian formalism was derived within classical mechanics for modelling a wide variety of physical systems. The temporal evolution of such systems is fully determined when the Hamiltonian function is known, and it is characterized by geometric properties such as the preservation of energy, the symplectic structure and the time-reversal symmetry of the flow [6, 7].

Numerical integrators that compute solutions preserving such properties are studied in the field of geometric numerical integration [7, 8]. Thus, deep learning, classical mechanics and geometric numerical integration are all relevant to the development of HNNs. In this work, we try to identify the optimal strategy for using numerical integrators when constructing loss functions for HNNs that are trained on noisy and sparse data.

Generally, we aim at learning autonomous systems of first-order ordinary differential equations (ODE)

$$\frac{d}{dt}y = f(y(t)), \quad y : [0, T] \to \mathbb{R}^n. \tag{1}$$

In the traditional setting, solving an initial value problem (IVP) means computing approximated solutions $y_n \approx y(t_n)$ when the vector field $f(y)$ and an initial value $y(t_0) = y_0$ are known. The focus of our study is the corresponding inverse problem; assuming knowledge of multiple noisy samples of the solution, $S_N = \{\tilde{y}_n\}_{n=0}^N$, the aim is to approximate the vector field $f$ with a neural

network model $f_\theta$. We will assume that the observations originate from a (canonical) Hamiltonian system, with a Hamiltonian $H : \mathbb{R}^{2d} \to \mathbb{R}$, where the vector field is given by

$$f(y) = J\nabla H(y(t)), \quad J := \begin{bmatrix} 0 & I \\ -I & 0 \end{bmatrix} \in \mathbb{R}^{2d \times 2d}. \tag{2}$$

This allows for learning the Hamiltonian function directly by setting $f_\theta(y) = J\nabla H_\theta(y)$, as proposed initially in [5].

Recently, many works highlight the benefit of using symplectic integrators when learning Hamiltonian neural networks [9, 10, 11, 12]. Here, we study what happens if, instead of using symplectic methods, efficient and higher-order MIRK methods are applied for inverse problems. We develop different approaches and apply them to learn highly oscillatory and chaotic dynamical systems from noisy data. The methods are general, they are not limited to separable Hamiltonian systems, and could indeed be used to learn any first-order ODE. However we focus our study on Hamiltonian systems, in order to build on the latest research on HNNs. Specifically, we compare our methods to the use of symplectic integrators to train Hamiltonian neural networks. Our contributions can be summarized as follows:

- We introduce the **mean inverse integrator** (MII), which efficiently averages trajectories of MIRK methods in order to increase accuracy when learning vector fields from noisy data (Definition 5.1).

- We present an **analysis of the sensitivity** of the loss function to perturbations giving insight into when the MII method yields improvement over a standard one-step scheme (Theorem 5.2).

- We show that **symplectic MIRK** methods have at most order $p = 2$ (Theorem 4.4). Particularly, the second-order implicit midpoint method is the symplectic MIRK method with minimal number of stages.

Finally, numerical experiments on several Hamiltonian systems benchmark MII against one-step training and symplectic recurrent neural networks (SRNN) [10], which rely on the Störmer–Verlet integrator. The structural difference between these three approaches is presented in Figure 2. Additionally, we demonstrate that substituting Störmer–Verlet with the classic Runge–Kutta method (RK4) in the SRNN framework yields significant reduction in error and allows accurate learning of non-separable Hamiltonian systems.

## 2  Related work

Hamiltonian neural networks was introduced in [5]. The numerical integration of Hamiltonian ODEs and the preservation of the symplectic structure of the ODE flow under numerical discretization have been widely studied over several decades [8, 7]. The symplecticity property is key and could inform the neural network architecture [13] or guide the choice of numerical integrator, yielding a theoretical guarantee that the learning target is actually a (modified) Hamiltonian vector field [14, 9], building on the backward error analysis framework [8]. Discrete gradients is an approach to numerical integration that guarantees exact preservation of the (learned) Hamiltonian, and an algorithm for training Hamiltonian neural networks using discrete gradient integrators is developed in [15] and extended to higher order in [16].

Since we for the inverse problem want to approximate the time-derivative of the solution, $f$, using only $\tilde{y}_n$, we need to use a numerical integrator when specifying the neural network loss function. For learning dynamical systems from data, explicit methods such as RK4 are much used [5, 17, 18]. However, explicit methods cannot in general preserve time-symmetry or symplecticity, and they often have worse stability properties compared to implicit methods [19]. Assuming that the underlying Hamiltonian is separable allows for explicit integration with the symplectic Störmer–Verlet method, which is exploited in [10, 20]. Symplecticity could be achieved without the limiting assumption of separability by training using the implicit midpoint method [12]. As pointed out in [12], this integrator could be turned into an explicit method in training by inserting sequential training data $\tilde{y}_n$ and $\tilde{y}_{n+1}$. In fact, the MIRK class [21, 22] contains all Runge–Kutta (RK) methods (including the midpoint method) that could be turned into explicit schemes when inserting the training data. This is exploited in [23], where high-order MIRK methods are used to train HNNs, achieving accurate

interpolation and extrapolation of a single trajectory with large step size, few samples and assuming zero noise.

The assumption of noise-free data limits the potential of learning from physical measurements or applications on data sets from industry. This issue is addressed in [10], presenting symplectic recurrent neural networks (SRNN). Here, Störmer–Verlet is used to integrate multiple steps and is combined with initial state optimization (ISO) before computing the loss. ISO is applied after training $f_\theta$ a given number of epochs and aims at finding the optimal initial value $\hat{y}_0$, such that the distance to the subsequent observed points $\tilde{y}_1, \ldots, \tilde{y}_N$ is minimized when integrating over $f_\theta$. While [10] is limited by only considering separable systems, [24] aims at identifying the optimal combination of third order polynomial basis functions to approximate a cubic non-separable Hamiltonian from noisy data, using a Bayesian framework.

## 3  Background on numerical integration

Some necessary and fundamental concepts on numerical integration and the geometry of Hamiltonian systems are presented below to inform the discussion on which integrators to use in inverse problems. Further details could be found in Appendix C.

**Fundamental concepts:** An important subclass of the general first-order ODEs (1) is the class of Hamiltonian systems, as given by (2). Often, the solution is partitioned into the coordinates $y(t) = [q(t), p(t)]^T$, with $q(t), p(t) \in \mathbb{R}^d$. A separable Hamiltonian system is one where the Hamiltonian could be written as the sum of two scalar functions, often representing the kinetic and potential energy, that depend only on $q$ and $p$ respectively, this means we have $H(q, p) = H_1(q) + H_2(p)$.

The $h$ flow of an ODE is a map $\varphi_{h,f} : \mathbb{R}^n \to \mathbb{R}^n$ sending an initial value $y(t_0)$ to the solution of the ODE at time $t_0 + h$, given by $\varphi_{h,f}(y(t_0)) := y(t_0 + h)$. A numerical integration method $\Phi_{h,f} : \mathbb{R}^n \to \mathbb{R}^n$ is a map approximating the exact flow of the ODE, so that

$$y(t_1) \approx y_1 = \Phi_{h,f}(y_0).$$

Here, $y(t_n)$ represents the exact solution and we denote with $y_n$ the approximation at time $t_n = t_0 + nh$. It should be noted that the flow map satisfies the following group property:

$$\varphi_{h_1,f} \circ \varphi_{h_2,f}\big(y(t_0)\big) = \varphi_{h_1,f}\big(y(t_0 + h_2)\big) = \varphi_{h_1+h_2,f}(y(t_0)). \tag{3}$$

In other words, a composition of two flows with step sizes $h_1, h_2$ is equivalent to the flow map over $f$ with step size $h_1 + h_2$. This property is not shared by numerical integrators for general vector fields. The order of a numerical integrator $\Phi_{h,f}$ characterizes how the error after one step depends on the step size $h$ and is given by the integer $p$ such that the following holds:

$$\|y_1 - y(t_0 + h)\| = \|\Phi_{h,f}(y_0) - \varphi_{h,f}(y(t_0))\| = \mathcal{O}(h^{p+1}).$$

**Mono-implicit Runge–Kutta methods:** Given vectors $b, v \in \mathbb{R}^s$ and a strictly lower triangular matrix $D \in \mathbb{R}^{s \times s}$, a MIRK method is a Runge–Kutta method where $A = D + vb^T$ [25, 26] and we assume that $[A]_{ij} = a_{ij}$ is the stage-coefficient matrix. This implies that the MIRK method can be written on the form

$$y_{n+1} = y_n + h \sum_{i=1}^{s} b_i k_i,$$

$$k_i = f\big(y_n + v_i(y_{n+1} - y_n) + h \sum_{j=1}^{s} d_{ij} k_j\big). \tag{4}$$

Specific MIRK methods and further details on Runge–Kutta schemes is discussed in Appendix C.2.

**Symplectic methods:** The flow map of a Hamiltonian system is symplectic, meaning that its Jacobian $\Upsilon_\varphi := \frac{\partial}{\partial y} \varphi_{h,f}(y)$ satisfies $\Upsilon_\varphi^T J \Upsilon_\varphi = J$, where $J$ is the same matrix as in (2). As explained in [8, Ch. VI.2], this is equivalent to the preservation of a projected area in the phase space of $[q, p]^T$. Similarly, a numerical integrator is symplectic if its Jacobian $\Upsilon_\Phi := \frac{\partial}{\partial y_n} \Phi_{h,f}(y_n)$ satisfies $\Upsilon_\Phi^T J \Upsilon_\Phi = J$. It is possible to prove [8, Ch. VI.4] that a Runge–Kutta method is symplectic if and only if the coeffients satisfy

$$b_i a_{ij} + b_j a_{ji} - b_i b_j = 0, \quad i, j = 1, \ldots, s. \tag{5}$$

## 4 Numerical integration schemes for solving inverse problems

We will now consider different ways to use numerical integrators when training Hamiltonian neural networks and present important properties of MIRK methods, a key component of the MII that is presented in Chapter 5.

**Inverse ODE problems in Hamiltonian form:** We assume to have potentially noisy samples $S_N = \{\tilde{y}\}_{n=0}^{N}$ of the solution of an ODE with vector field $f$. The inverse problem can be formulated as the following optimization problem:

$$\arg\min_{\theta} \sum_{n=0}^{N-1} \left\| \tilde{y}_{n+1} - \Phi_{h,f_\theta}(\tilde{y}_n) \right\|, \tag{6}$$

where $f_\theta = J\nabla H_\theta$ is a neural network approximation with parameters $\theta$ of a Hamiltonian vector field $f$, and $\Phi_{h,f_\theta}$ is a one-step integration method with step length $h$. In the setting of inverse ODE problems, the availability of sequential points $S_N$ could be exploited when a numerical method is used to form interpolation conditions, for $f_\theta \approx f$ for each $n$ in the optimization problem (6). For example, $\tilde{y}_n$ and $\tilde{y}_{n+1}$ could be inserted in the implicit midpoint method, turning a method that is implicit for IVPs into an explicit method for inverse problems:

$$\Phi_{h,f_\theta}(\tilde{y}_n, \tilde{y}_{n+1}) = \tilde{y}_n + hf_\theta\left(\frac{\tilde{y}_n + \tilde{y}_{n+1}}{2}\right). \tag{7}$$

We denote this as the inverse injection, which defines an inverse explicit property for numerical integrators.

**Definition 4.1** (Inverse injection). Assume that $\tilde{y}_n, \tilde{y}_{n+1} \in S_N$. Let the *inverse injection* for the integrator $\Phi_{h,f}(y_n, y_{n+1})$ be given by the substitution $(\tilde{y}_n, \tilde{y}_{n+1}) \rightarrow (y_n, y_{n+1})$ such that

$$\hat{y}_{n+1} = \Phi_{h,f}(\tilde{y}_n, \tilde{y}_{n+1}).$$

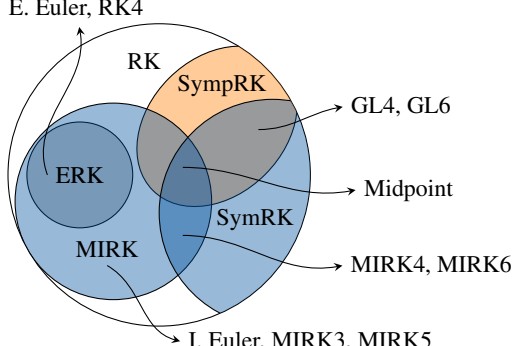

Figure 1: Venn diagram of Runge–Kutta (RK) subclasses: explicit RK (ERK), symplectic RK (SympRK), mono-implicit RK (MIRK) and symmetric RK (SymRK).

**Definition 4.2** (Inverse explicit). A numerical one-step method $\Phi$ is called *inverse explicit* if it is explicit under the inverse injection.

This procedure is utilized successfully by several authors when learning dynamical systems from data, see e.g. [12, 27]. However, this work is the first attempt at systematically exploring numerical integrators under the inverse injection, by identifying the MIRK methods as the class consisting of inverse explicit Runge–Kutta methods.

**Proposition 4.3.** *MIRK-methods are inverse explicit.*

*Proof.* Since the matrix $D$ in (4) is strictly lower triangular, the stages are given by

$$\begin{aligned}
k_1 &= f(y_n + v_i(y_{n+1} - y_n)) \\
k_2 &= f(y_n + v_i(y_{n+1} - y_n) + hd_{21}k_1) \\
&\vdots \\
k_s &= f\left(y_n + v_i(y_{n+1} - y_n) + h\sum_{j=1}^{s-1} d_{sj}k_j\right)
\end{aligned}$$

meaning that if $y_n$ and $y_{n+1}$ are known, all stages, and thus the next step $\hat{y}_{n+1} = y_n + h\sum_{i=1}^{s} b_i k_i$, could be computed explicitly. □

Because of their explicit nature when applied to inverse ODE problems, MIRK methods are an attractive alternative to explicit Runge–Kutta methods; in contrast to explicit RK methods, they can be symplectic or symmetric, or both, without requiring the solution of systems of nonlinear

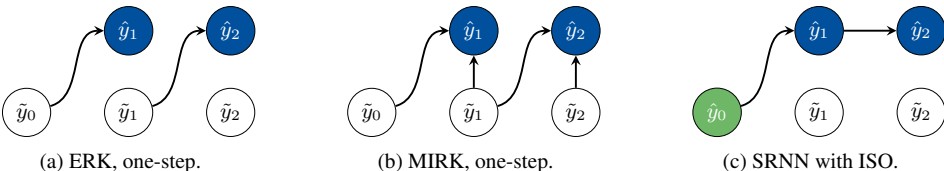

Figure 2: Differences of observation dependency, assuming $N = 2$ for explicit and mono-implicit one-step training, and explicit multi-step training with initial state optimization (green node $\hat{y}_0$).

equations, even when the Hamiltonian is non-separable. Figure 1 illustrates the relation between various subclasses and the specific methods are described in Table 1 in Appendix C. In addition, for $s$-stage MIRK methods, it is possible to construct methods of order $p = s + 1$ [22]. This is in general higher order than what is possible to obtain with $s$-stage explicit Runge–Kutta methods. Further computational gains could also be made by reusing evaluations of the vector field between multiple steps, which using MIRK methods allow for, as explained in Appendix I. The dependency structure on the data $S_N$ of explicit RK (ERK) methods, MIRK methods and the SRNN method [10] is illustrated in Figure 2.

**Maximal order of symplectic MIRK methods:** From the preceding discussion, it is clear that symplectic MIRK methods are of interest when learning Hamiltonian systems from data, since they combine computational efficiency with the ability to preserve useful, geometric properties. Indeed, symplectic integrators in the training of HNNs have been considered in [9, 10, 11, 12, 13]. The subclass of symplectic MIRK methods is represented by the middle, dark blue field in the Venn diagram of Figure 1. The next result gives an order barrier for symplectic MIRK methods that was, to the best of our knowledge, not known up to this point.

**Theorem 4.4.** *The maximum order of a symplectic MIRK method is* $p = 2$.

*Proof.* This is a shortened version of the full proof, which can be found in Appendix F. A MIRK method is a Runge–Kutta method with coefficients $a_{ij} = d_{ij} + v_i b_j$. Requiring $d_{ij}, b_i$ and $v_i$ to satisfy the symplecticity conditions of (5) in addition to $D$ being strictly lower triangular, yields the following restrictions

$$
\begin{aligned}
b_i d_{ij} + b_i b_j (v_j + v_i - 1) = 0, \quad &\text{if } i \neq j, \\
b_i = 0 \ \text{ or } \ v_i = \frac{1}{2}, \quad &\text{if } i = j, \\
d_{ij} = 0, \quad &\text{if } i > j.
\end{aligned}
\tag{8}
$$

These restrictions result in an RK method that could be reduced to choosing a coefficient vector $b \in \mathbb{R}^s$ and choosing stages on the form $k_i = f\left(y_n + \frac{h}{2} \sum_j^s b_j k_j\right)$ for $i = 1, \ldots, s$. It is then trivial to check that this method can only be of up to order $p = 2$. Note that for $s = 1$ and $b_1 = 1$ we get the midpoint method. $\square$

**Numerical integrators outside the RK class:** While this paper is mainly concerned with MIRK methods, several other types of numerical integrators could be of interest for inverse problems. *Partitioned Runge–Kutta methods* are an extension and not a subclass of RK methods, and can be symplectic and symmetric, while also being explicit for separable Hamiltonian systems. The Störmer–Verlet integrator of order $p = 2$ is one example. Higher order methods of this type are derived in [28] and used for learning Hamiltonian systems in [29, 30]. *Discrete gradient methods* [31, 32] are inverse explicit and well suited to train Hamiltonian neural networks using a modified automatic differentiation algorithm [15]. This method could be extended to higher order methods as shown in [16]. In contrast to symplectic methods, discrete gradient methods preserve the Hamiltonian exactly up to machine precision. A third option is *elementary differential Runge–Kutta methods* [33], where for instance [34] show how to use backward error analysis to construct higher order methods from modifications to the midpoint method. This topic is discussed further in Appendix H, where we also present a novel, symmetric discrete gradient method of order $p = 4$.

## 5 Mean inverse integrator for handling noisy data

**Noisy ODE sample:** It is often the case that the samples $S_N$ are not exact measurements of the system, but perturbed by noise. In this paper, we model the noise as independent, normally distributed

perturbations

$$\tilde{y}_n = y(t_n) + \delta_n, \quad \delta_n \sim \mathcal{N}(0, \sigma^2 I), \tag{9}$$

where $\mathcal{N}(0, \sigma^2 I)$ represents the multivariate normal distribution. With this assumption, a standard result from statistics tells us that the variance of a sample-mean estimator with $N$ samples converges to zero at the rate of $\frac{1}{N}$. That is, assuming that we have $N$ samples $\tilde{y}_n^{(1)}, \ldots, \tilde{y}_n^{(N)}$, then

$$\mathrm{Var}[\overline{y}_n] = \mathrm{Var}\left[\frac{1}{N} \sum_{j=1}^{N} \tilde{y}_n^{(j)}\right] = \frac{\sigma^2}{N}.$$

Using the inverse injection with the midpoint method, the vector field is evaluated in the average of $\tilde{y}_n$ and $\tilde{y}_{n+1}$, reducing the variance of the perturbation by a factor of two, compared to evaluating the vector field in $\tilde{y}_n$, as is done in all explicit RK methods. Furthermore, considering the whole data trajectory $S_N$, multiple independent approximations to the same point $y(t_n)$ can enable an even more accurate estimate. This is demonstrated in the analysis presented in Theorem 5.2 and in Figure 4.

**Averaging multiple trajectories:** In the inverse ODE problem, we assume that there exists an *exact* vector field $f$ whose flow interpolates the discrete trajectories $S_N$, and the flow of this vector field satisfies the group property (3). The numerical flow $\Phi_{h,f}$ for a method of order $p$ satisfies this property only up to an error $\mathcal{O}(h^{p+1})$ over one step. In the presence of noisy data, compositions of one-step methods can be used to obtain multiple different approximations to the same point $y(t_n)$, by following the numerical flow from different nearby initial values $\tilde{y}_j, j \neq n$, and thus reduce the noise by averaging over these multiple approximations. Accumulation of the local truncation error is expected when relying on points further away from $t_n$. However, for sufficiently small step sizes $h$ compared to the size of the noise $\sigma$, one can expect increased accuracy when averaging over multiple noisy samples.

As an example, assume that we know the points $\{\tilde{y}_0, \tilde{y}_1, \tilde{y}_2, \tilde{y}_3\}$. Then $y(t_2)$ can be approximated by computing the mean of the numerical flows $\Phi_{h,f}$ starting from different initial values:

$$\begin{aligned}
\overline{y}_2 &= \frac{1}{3}\big(\Phi_{h,f}(\tilde{y}_1) + \Phi_{h,f} \circ \Phi_{h,f}(\tilde{y}_0) + \Phi_{-h,f}^*(\tilde{y}_3)\big) \\
&\approx \frac{1}{3}\big(\tilde{y}_0 + \tilde{y}_1 + \tilde{y}_3 + h(\Psi_{0,1} + 2\Psi_{1,2} - \Psi_{2,3})\big),
\end{aligned} \tag{10}$$

where we by $\Phi^*$ mean the adjoint method of $\Phi$, as defined in [8, Ch. V], and we let $\Psi_{n,n+1}$ be the increment of an inverse-explicit numerical integrator, so that

$$\Phi_{h,f}(\tilde{y}_n, \tilde{y}_{n+1}) = \tilde{y}_n + h\Psi_{n,n+1}.$$

For example, for the midpoint method, we have that $\Psi_{n,n+1} = f(\frac{\tilde{y}_n + \tilde{y}_{n+1}}{2})$. When stepping in negative time in (10), we use the adjoint method in order to minimize the number of vector field evaluations, also when non-symmetric methods are used (which implies that we always use e.g. $\Psi_{1,2}$ and not $\Psi_{2,1}$). Note that in order to derive the approximation in (10), repeated use of the inverse injection allows the known points $\tilde{y}_n$ to form an explicit integration procedure, where composition of integration steps are approximated by summation over increments $\Psi_{n,n+1}$. This approximation procedure is presented in greater detail in Appendix D.

**Mean inverse integrator:** The mean approximation over the whole trajectory $\overline{y}_n$, for $n = 0, \ldots, N$, could be computed simultaneously, reusing multiple vector field evaluations in an efficient manner. This leads to what we call the mean inverse integrator. For example, when $N = 3$ we get

$$\begin{bmatrix} \overline{y}_0 \\ \overline{y}_1 \\ \overline{y}_2 \\ \overline{y}_3 \end{bmatrix} = \frac{1}{3}\begin{bmatrix} 0 & 1 & 1 & 1 \\ 1 & 0 & 1 & 1 \\ 1 & 1 & 0 & 1 \\ 1 & 1 & 1 & 0 \end{bmatrix}\begin{bmatrix} \tilde{y}_0 \\ \tilde{y}_1 \\ \tilde{y}_2 \\ \tilde{y}_3 \end{bmatrix} + \frac{h}{3}\begin{bmatrix} -3 & -2 & -1 \\ 1 & -2 & -1 \\ 1 & 2 & -1 \\ 1 & 2 & 3 \end{bmatrix}\begin{bmatrix} \Psi_{0,1} \\ \Psi_{1,2} \\ \Psi_{2,3} \end{bmatrix},$$

and the same structure is illustrated in Figure 3.

**Definition 5.1** (Mean inverse integrator). For a sample $S_N$ and an inverse-explicit integrator $\Psi_{n,n+1}$, the mean inverse integrator is given by

$$\overline{Y} = \frac{1}{N}\left(U\tilde{Y} + hW\Psi\right) \tag{11}$$

where $\tilde{Y} := [\tilde{y}_0, \ldots, \tilde{y}_N]^T \in \mathbb{R}^{(N+1) \times m}$, $\Psi := [\Psi_{0,1}, \ldots, \Psi_{N-1,N}]^T \in \mathbb{R}^{N \times m}$.

Finally, $U \in \mathbb{R}^{(N+1) \times (N+1)}$ and $W \in \mathbb{R}^{(N+1) \times N}$ are given by

$$[U]_{ij} := \begin{cases} 0 & \text{if} \quad i = j \\ 1 & \text{else} \end{cases} \qquad \text{and} \qquad [W]_{ij} := \begin{cases} j - 1 - N & \text{if} \quad j \geq i \\ j & \text{else} \end{cases}.$$

By substituting the known vector field $f$ with a neural network $f_\theta$ and denoting the matrix containing vector field evaluations by $\Psi_\theta$ such that $\overline{Y}_\theta := \frac{1}{N}(U\tilde{Y} + hW\Psi_\theta)$, we can formulate an analogue to the inverse problem (6) by

$$\arg\min_\theta \left\| \tilde{Y} - \overline{Y}_\theta \right\|. \tag{12}$$

**Analysis of sensitivity to noise:** Consider the optimization problems using integrators either as one-step methods or MII by (6) resp. (12). We want to investigate how uncertainty in the data $\tilde{y}_n$ introduces uncertainty in the optimization problem. Assume, for the purpose of analysis, that the underlying vector field $f(y)$ is known. Let

$$\mathcal{T}_n^{\text{OS}} := \tilde{y}_n - \Phi_{h,f}(\tilde{y}_{n-1}, \tilde{y}_n),$$
$$\mathcal{T}_n^{\text{MII}} := \tilde{y}_n - [\overline{Y}]_n$$

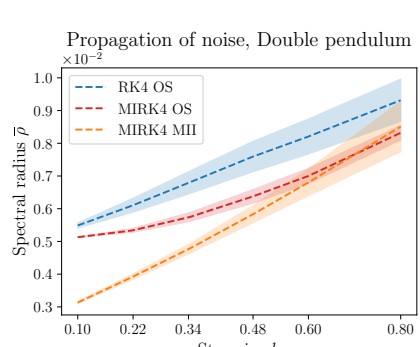

Figure 3: Illustration of the structure of the mean inverse integrator for $N = 3$.

be the *optimization target* or the expression one aims to minimize using a one-step method (OS) and the MII, where $\overline{Y}$ is given by Definition 5.1. For a matrix $A$ with eigenvalues $\lambda_i(A)$, the spectral radius is given by $\rho(A) := \max_i |\lambda_i(A)|$. An analytic expression that approximates $\rho(\mathcal{T}_n^{\text{OS}})$ and $\rho(\mathcal{T}_n^{\text{MII}})$ by linearization of $f$ for a general MIRK method is provided below.

**Theorem 5.2.** *Let $S_N = \{\tilde{y}_n\}_{n=0}^N$ be a set of noisy samples, equidistant in time with step size $h$, with Gaussian perturbations as defined by (9) with variance $\sigma^2$. Assume that a MIRK integrator $\Phi_{h,f}$ is used as a one-step method. Then the spectral radius is approximated by*

$$\rho_n^{OS} := \rho\left( Var\left[\mathcal{T}_n^{OS}\right] \right) \approx \sigma^2 \left\| 2I + hb^T(\mathbb{1} - 2v)(f' + f'^T) + h^2 Q^{OS} \right\|_2, \tag{13}$$

$$\rho_n^{MII} := \rho\left( Var\left[\mathcal{T}_n^{MII}\right] \right) \approx \frac{\sigma^2}{N} \left\| (1 + N)I + hP_{nn} + \frac{h}{N}\sum_{\substack{j=0 \\ j \neq n}}^s P_{nj} + \frac{h^2}{N}Q^{MII} \right\|_2, \tag{14}$$

*where $f' := f'(y_n)$ and $P_{nj}, Q^{OS}$ and $Q^{MII}$ (defined in (24) in Appendix G) are matrices independent of the step size $h$.*

The proof is found in Appendix G. Let $\alpha := b^T(\mathbb{1} - 2v)$ denote the coefficients of the first order term in $h$ of Equation (13). For any explicit RK method we have that $v = 0$ and since $b^T\mathbb{1} = 1$ (method of at least order one) we find that $\alpha_{\text{ERK}} = 1$. Considering the Butcher tableau of MIRK4 in Figure 9 we find that $\alpha_{\text{MIRK4}} = 0$. Thus, as $h \to 0$ we would expect quadratic convergence of MIRK4 and linear convergence of RK4 for $\rho_n^{\text{OS}}$ to $2\sigma^2$. Considering MII (14) one would expect linear convergence for $\rho_n^{\text{MII}}$ to $\sigma^2$ if $N$ is large, as $h \to 0$.

A numerical approximation of $\rho_n^{\text{OS}}$ and $\rho_n^{\text{MII}}$ could be realized by a Monte-Carlo estimate. We compute the spectral radius $\hat{\rho}_n$ of the empirical covariance matrix of $\mathcal{T}_n^{\text{OS}}$ and $\mathcal{T}_n^{\text{MII}}$ by sampling $5 \cdot 10^3$ normally distributed perturbations $\delta_n$ with $\sigma^2 = 2.5 \cdot 10^{-3}$ to each point $y_n$ in a trajectory of $N + 1$ points and step size $h$. We then compute the

Figure 4: Average of $\overline{\rho}$ over 10 trajectories. Shaded area represent one standard deviation.

270 trajectory average $\overline{\rho} = \frac{1}{N+1}\sum_{n=0}^{N}\hat{\rho}_n$, fix the end time $T = 2.4$, repeat the approximations for
271 decreasing step sizes $h$ and increasing $N$ and compute the average of $\overline{\rho}$ for 10 randomly sampled
272 trajectories $S_N$ from the double pendulum system. The plot in Figure 4 corresponds well with what
273 one would expect from Theorem 5.2 and confirms that first MIRK (with $v \neq 0$) and secondly MII
274 reduces the sensitivity to noise in the optimization target.

## 6 Experiments

**Methods and test problems:** We train HNNs us-
ing different integrators and methods in the inverse
problem (6). We use MIRK4 together with the MII
method and compare to the implicit midpoint method,
RK4 and MIRK4 applied as one-step methods, as
well as ISO followed by Störmer–Verlet and RK4
integrated over multiple time-steps. The latter strat-
egy, illustrated in Figure 2, was suggested in [10],
where Störmer–Verlet is used. Separable networks
$H_\theta(q, p) = H_{1,\theta}(q) + H_{2,\theta}(p)$ are trained on data
from the Fermi–Pasta–Ulam–Tsingou (FPUT) prob-
lem and the Hénon–Heiles system. For the double
pendulum, which is non-separable, a fully connected
network is used for all methods except Störmer–
Verlet, which requires separability in order to be ex-
plicit. The Hamiltonians are described in Appendix
A and all systems have solutions $y(t) \in \mathbb{R}^4$.

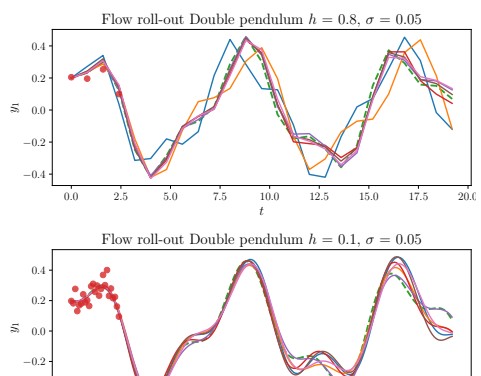

Figure 5: Roll-out in time obtained by inte-
grating over the learned vector fields when
training on data from the double pendulum
Hamiltonian.

After using the specified integrators in training, ap-
proximated solutions are computed for each learned
vector field $f_\theta$ using the Scikit-learn implementation
of DOP853 [35], which is also used to generate the
training data. The error is averaged over $M = 10$
points and we find what we call the flow error by

$$e(f_\theta) = \frac{1}{M}\sum_{n=1}^{M}\|\hat{y}_n - y(t_n)\|_2, \quad y(t_n) \in S_M^{\text{test}},$$
$$\hat{y}_{n+1} = \Phi_{h,f_\theta}(y_n). \tag{15}$$

**Training data:** Training data is generated by sampling $N_2 = 300$ random initial values $y_0$ requiring
that $0.3 \le \|y_0\|_2 \le 0.6$. The data $S_{N_1,N_2} = \{\tilde{y}_n^{(j)}\}_{n=0,j=0}^{N_1,N_2}$ is found by integrating the initial values
with DOP853 with a tolerance of $10^{-15}$ for the following step sizes and number of steps: $(h, N_1) =$
$(0.4, 4), (0.2, 8), (0.1, 16)$. The points in the flow are perturbed by noise where $\sigma \in \{0, 0.05\}$. Error
is measured in $M = 10$ random points in the flow, within the same domain as the initial values.
Furthermore, experiments are repeated with a new random seed for the generation of data and
initialization of neural network parameters five times in order to compute the standard deviation of
the flow error. The flow error is shown in Figure 6. Additional results are presented in Appendix B.

**Neural network architecture and optimization:** For all test problems, the neural networks have 3
layers with a width of 200 neurons and $\tanh(\cdot)$ as the activation function. The algorithms are imple-
mented using PyTorch [36] and the code for performing ISO is a modification of the implementation
by [10][1]. Training is done using the quasi-Newton L-BFGS algorithm [37] for 20 epochs without
batching. This optimization algorithm is often used to train physics-informed neural networks [1]
and in this setting it proved to yield superior results in comparison to the often used Adam optimizer.
Further details are provided in Appendix E.

**Results:** As observed in Figure 6 and supported by the analytical result illustrated in Figure 4, the MII
approach facilitates more accurate training from from noisy data than one-step methods. However,
training with multiple integration steps in combination with ISO yields lower error when RK4 is used

[1]https://github.com/zhengdao-chen/SRNN (CC-BY-NC 4.0 License)

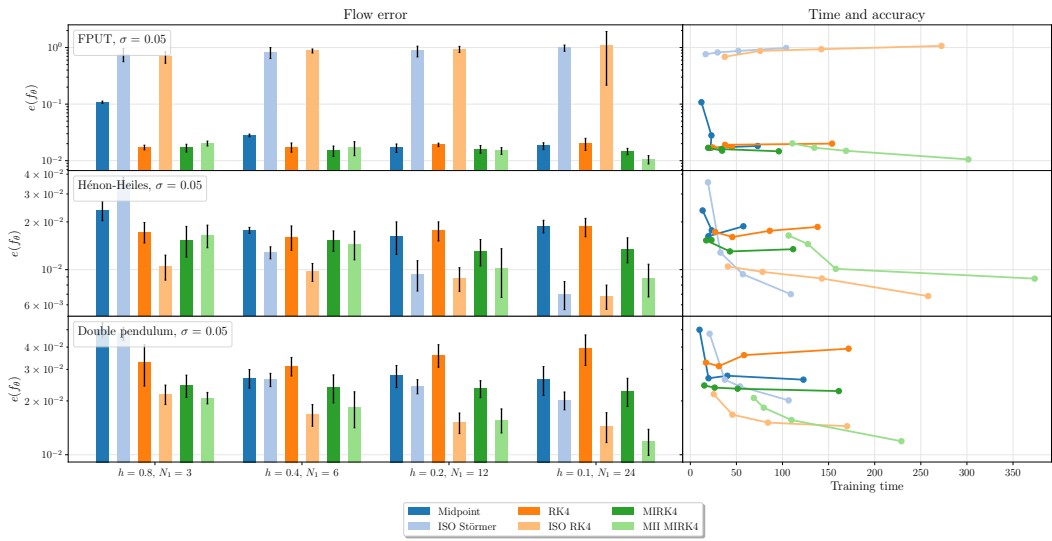

Figure 6: The flow error when learning vector fields using one-step methods directly (Midpoint, RK4 and MIRK4), ISO and multiple time-steps (ISO Störmer and ISO RK4) and MII (MII MIRK4). The error bars display the standard deviation after rerunning 5 experiments on data with $\sigma = 0.05$. The right subplot shows the computational time used in training against the flow error.

for the Hénon–Heiles problem and similar performance as MII on the double pendulum. We notice that the SRNN approach, i.e. ISO with Störmer–Verlet, is improved when switching to RK4, which means sacrificing symplecticity to achieve higher order. The results for FPUT stand out in Figure 6, since both ISO methods have large errors here. The roll-out in time of the learned vector fields is presented in Figure 8 in Appendix B, where the same can be observed. As also could be seen here, the FPUT Hamiltonian gives rise to highly oscillatory trajectories, and the errors observed in Figure 6 might indicate that ISO is ill-suited for this kind of dynamical systems.

Two observations could be made regarding the one-step methods without averaging or ISO. First, it is likely that the midpoint method has weaker performance for large step sizes due to its lower order, compared to both RK4 and MIRK4, despite the fact that it is a symplectic method. The same is clear from Figure 7 in Appendix B, which display the flow error when training on data without noise. Secondly, building on the sensitivity analysis, we observe that MIRK4 consistently attains higher accuracy than RK4, as expected from the Monte-Carlo simulation found in Figure 4.

## 7   Conclusion

In this work we present the mean inverse integrator, which allows both chaotic and oscillatory dynamical systems to be learned with high accuracy from noisy data. Within this method, integrators of the MIRK class are a key component. To analyse how noise is propagated when training with MII and MIRK, compared to much used explicit methods such as RK4, we developed a sensitivity analysis that is verified both by a Monte-Carlo approximation and reflected in the error of the learned vector fields. Finally, we build on the SRNN [10] by replacing Störmer–Verlet with RK4, and observer increased performance. When also considering the weak performance of the implicit midpoint method, this tells us that order might be of greater importance than preserving the symplectic structure when training HNNs. Both the MIRK methods, the mean inverse integrator and initial state optimization form building blocks that could be combined to form novel approaches for solving inverse problems and learning from noisy data.

**Limitations:** The experiments presented here assume that both the generalized coordinates $q_n$ and the generalized momenta $p_n$ could be observed. In a setting where HNNs are to model real and not simulated data, the observations might lack generalized momenta [38] or follow Cartesian coordinates, requiring the enforcement of constraints [17, 39]. Combining approaches that are suitable for data that is both noisy and follow less trivial coordinate systems is a subject for future research.

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
