# OpenReview forum: "Learning Dynamical Systems from Noisy Data with Inverse-Explicit Integrators"
_NeurIPS.cc/2023/Conference — Submitted to NeurIPS 2023_

### Official Review · Reviewer_yxuC · 2023-07-03

**Soundness:** 2 fair
**Presentation:** 2 fair
**Contribution:** 3 good
**Rating:** 5
**Confidence:** 2

**Summary:**

This paper introduces a new integration method (mean inverse integrator) for learning dynamics from noisy data. Experiments on Hamiltonian systems show the effectiveness of the proposed method.

**Strengths:**

- The problem of learning physical dynamics from noisy data is an interesting one.
- It combines techniques from the field of numerical computation with machine learning.


**Weaknesses:**

- The usefulness and significance of the proposed method is not clear.
- I feel that comparative experiments are not sufficient.
- It is not clear what advantages the proposed method has over the naive noise handling method.

**Questions:**

- To handle noisy data, the most common method is to introduce an observation model (e.g., Normal distribution) and learn the noise variance by maximum likelihood estimation. Can you tell us what advantages the proposed method has over such naive methods?

- It would be good to clarify situations that can only be solved by the proposed method and emphasize the significance of this paper. For example, when using Gaussian likelihood, the noise variance is usually assumed to be constant. Can the proposed method relax this assumption? In other words, can the proposed method be applied to cases where the noise variance depends on time and state?

- A thorough comparison with the latest methods (e.g., [9-12, 15]) would be helpful.

- The font size in Fig. 5 and 6 is too small and should be enlarged.

**Limitations:**

It would be good to add a careful discussion of the advantages and disadvantages of the proposed method.

---

> ### Author Rebuttal · Authors · 2023-08-09
>
> We are grateful for the helpful comments and suggestions provided by the reviewer. Below are our responses.
>
> ### Assumptions on noise (Q1 - Q2)
>
> - The reviewer points to one of the advantages of MII, which we will make more clear in the revised version: that it puts no assumption on the noise / data distribution. In the paper we assume Gaussian noise, to make the analysis used to derive Theorem 5.2 more straightforward. However, the method itself is derived by exploiting the fundamental group property of the underlying flow of the exact solution (and approximating this in discrete time), allowing multiple independent approximations to be produced. Hence, the method is derived from a numerical analysis perspective, more than a statistics / Bayesian perspective.
> - As the reviewer alludes to, the MII is thus distribution agnostic. The only underlying assumption is the group property of the flow map, namely that $\varphi_{h_1} \circ \varphi_{h_2} (y_0) = \varphi_{h_1 + h_2}  (y_0)$. This is a very general property of dynamical systems and the MII is thus applicable to a wide range of problems. We will strive to include numerical experiments with time- and state-dependent noise in the final version of the paper and thank the reviewer for the suggestion.
>
> ### Comparison with other methods (Q3)
> (References in this section are to the references in the paper)
>
> - Chen et al. [10] introduce the method that is called Störmer + ISO in our paper, and is thus included in the experiments. We also extend on [10] by replacing Störmer-Verlet with RK$4$ and show that this is in many cases superior, even though it is not a symplectic scheme.
> - Zhu et al. [11] apply inverse modified equations to provide an existence guarantee for Hamiltonian neural networks when the integrator is symplectic. The midpoint method is symplectic and thus included in the experiments. We have also included results from the symplectic Gauss–Legendre methods of order $4$ and $6$ in Figure 3.
> - The works of David and Méhats [12] and Offen and Ober-Blöbaum [9] are similar to [11] in how they argue for symplectic methods, but take this a step further by exploiting the inverse modified equation to derive a correction term to the learned Hamiltonian. This approach is theoretically appealing and we will include a comparison with this method in the final version of the paper.
> - Matsubara et al. [15] uses discrete gradient methods and exploits the discrete chain rule to derive a novel automatic differentiation algorithm for computing discrete gradients of neural networks. We have included a fourth-order method (DGM$4$) that builds on this work in the experiment in Figure 3. We have run multiple tests using discrete gradient methods of order $2$ and $4$ and find that they do not improve upon symplectic methods of the same order when there is no noise, and they are also not as robust to noise.
>
> Figure 3 also includes experiments with the modified implicit midpoint method (MIMP$4$) proposed in the appendix. Neither of these methods differ significantly in accuracy from the methods used for experiments in the paper as it is.
>
> ### Other comments
>
> - We will increase the font size in figures 5 and 6 in the final version.
> - With the option to add one more page in the revised version, we will include an extended and improved discussion of the advantages and disadvantages of the proposed method, following along the lines of the comments above.

---

> > ### Comment · Reviewer_yxuC · 2023-08-18
> > **Reply**
> >
> > Thank you for your answer. I now understand the advantage of the proposed method (no need to assume noise distribution). I will raise my score by 1.

---

### Official Review · Reviewer_HMrx · 2023-07-06

**Soundness:** 3 good
**Presentation:** 2 fair
**Contribution:** 3 good
**Rating:** 6
**Confidence:** 4

**Summary:**

The paper investigates mono-implicit Runge--Kutta (MIRK) methods for learning dynamical systems from data. In particular, MIRK methods can be made explicit by introducing the external data into the solver step itself, leading to a more efficient integrator while keeping favorable stability, symmetry, and symplecticity properties. To handle noisy data, the paper proposes the "mean inverse integrator" as an efficient way to average multiple trajectories and learn meaningful vector fields from these. The methods are demonstrated in multiple numerical experiments.

**Strengths:**

The paper is well-written, the proposed method is presented very clearly, and the main claims made are well-supported.

**Weaknesses:**

I find the presentation of the explicit Runge--Kutta (ERK) baseline a bit confusing, and in particular counter-intuitive of my current understanding of the usage of ERK methods for inverse problems in the context of ODEs. Concrete questions are below, in "Questions".
This criticism also extends to insufficiently detailed baselines, and potentially to insufficient baselines overall.

**Questions:**

See also "Weaknesses". Notably:
1. How exactly is the RK4 / ISO+RK4 baseline defined?
2. Why is (6) _the_ appropriate loss function? I would rather assume something of the type $\sum || \tilde{y}_n - \hat{y}_n ||$, where $\hat{y}_n$ is a state estimate computed by the method; for instance by applying a numerical solver to simulate a whole trajectory. In particular, extrapolating exactly from the last data point seems to be a strong restriction, in particular if the data point can be very noisy. I would not consider this "the inverse problem formulation", but rather a specific choice that is made, geared towards the specific algorithm of interest.
3. Related to the last question: Could you commend on the standard least-squares approach of integrating the whole trajectory from some initial value (estimate) with RK, as opposed to the described method of just integrating from one data point to the next? The latter seems to be related to "multiple shooting"-type approaches which, while they can be very beneficial, seems to be far from the standard for neural ODEs. Though, to the best of my knowledge, even in multiple shooting it is common to extrapolate some state estimates $\hat{y}$; though part of the objective is the closeness of these estimates to the data, and they are initialized on the data.

Other remarks:
- l. 201: "evaluationg the vector filed in $\tilde{y}_n$, as is done in all explicit RK methods": To the best of my understanding this is not done in all explicit RK methods, but only in multiple shooting-type approaches (and even there this might not hold). A standard least-squares approach simulates a whole trajectory and then computes an L2-loss---and thus does not need to evaluate $f$ on the data points themselves.
- l. 129: $\tilde{y}$ is missing an index
- Related work: The write-up is very helpful; but I think it could be further improved by highlighting better how the proposed paper relates to these methods and highlight similarities and differences.
- l. 72-73: I agree with the sentiment of the statement, but I think the strict _necessity for numerical integrators_ is somewhat disputed in the literature. Notably, "gradient matching" methods compute first an interpolant, and then adjust the vector field to match the known derivative of the interpolant---this circumvents using RK or any related numerical method, which is why such methods often claim to be "simulation-free".

**Limitations:**

The authors address limitations in a dedicated section, which is much appreciated.

One additional limitation that is not explicitly mentioned in the section that comes to mind is the influence of the noise on the data: Since the data is explicitly included into the numerical solver, as opposed to being just part of some L2 loss to guide some estimated trajectory, I would expect that for very noisy data other methods might be preferrable (in particular a least-squares approach with RK from a learned initial value).

Another potential limitation: I assume that this method is not able to be a plug-in replacement in some latent neural ODE setting, where the ODE trajectory is not observed directly but only in some transformed space, e.g. when having video data and modeling an ODE in latent space? This is of course far from a trivial setting and I do not expect that such specific scenarios need to be mentioned explicitly; but the necessity of actual trajectory observations, as opposed to partial or non-linear observations, could indeed be another limitation.

---

> ### Author Rebuttal · Authors · 2023-08-09
>
> We thank the reviewer for the insightful comments, which we will use to improve our paper in its revised version. Below are our responses.
>
> ### Our method and baselines, and relations to other methods (Q1 - Q3)
>
> The RK$4$ baseline is defined as in Eq. 6, meaning that the vector field is trained taking only one step with the numerical integrator. ISO + RK$4$ on the other hand is of the recurrent type, meaning a whole trajectory approximation is produced, integrating a full sequence $\hat y_n$ where we have first found the optimal $\hat y_0$ producing the trajectory of minimal distance to the proceeding $\tilde y_n$. Then the loss is computed using $\sum \\|\tilde y_n - \hat y_n \\|$, which is the loss mentioned by the reviewer and referred to as the least-squares approach.
>
> Three comments should be made:
>
> 1. Computing the loss one step at a time is exactly what allows the MIRK class of integrators to be used explicitly with the inverse injection, as explained in Section 4 of the paper. For computing a full trajectory approximation with an implicit RK method, one would have to solve non-linear systems of equations in every step. Even though our approach limits the loss to be computed one step at a time, it allows symmetric, symplectic and stable integrators to be efficiently used in training.
>
> 2. The one-step approach is sensitive to noise, which is why we introduce MII to build an averaged approximation using linear combinations of the one-step approximations. This is the main contribution of the paper, and the approach is compared against what would be close to a least-squares approach in the ISO + RK$4$ baseline.
>
> 3. The least squares approach (both ISO methods) breaks down for the highly oscillatory FPUT problem, where MII produces high accuracy with a minimal increase in computational cost. A similar observation supporting multiple-shooting when the data is highly oscillatory is made in [1]. This demonstrates that MII is a more robust method for learning dynamics from noisy data.
>
> We agree with the reviewer that this difference should be made more clear in the paper (rather than defining (6) as the default approach). In our revised version we will do this. We will also put a greater emphasis on the benefit of MII for highly oscillatory problems.
>
> ### Replies to the other remarks
>
> - Regarding the first remark, about the sentence starting on line 201: We will reformulate this to make clear our meaning: that with our multiple-shooting type approach, using an explicit RK method means evaluating in $\tilde{y}_n$, which means that this class of integrators has a disadvantage in our case.
> - Remark II: We will add the index.
> - We will take remark III into account and improve the related work section. See our reply to reviewer yxuC for a more detailed comparison to the related work, which we will incorporate into the revision.
> - Regarding the fourth remark: We will rework the specific sentence, since we agree that it is at best imprecise the way it stands now. Moreover, we thank the reviewer for pointing to the gradient matching approach, since we see that our paper would benefit from mentioning this and how it relates to our approach: they share the advantage that integration does not have to be done during training. Specifically, we will note the equivalence between our approach and gradient matching for certain choices. E.g., using linear interpolation in gradient matching could be equivalent to using our approach with the one-step method and either the explicit Euler method or the implicit midpoint method, depending on where the loss is evaluated.
>
>
> ### Limitations
>
> - That a method based on integrating over several steps from a learned initial value might be better for very noisy data is a deft assumption. Indeed, we have performed numerical experiments that support this; see Figure 2 in the attached PDF, where ISO+RK$4$ is a method of this type and does indeed outperform the MII approach. We will include this result in the revision and comment on it in the limitations section.
> - It is true that the method would not be a plug-in replacement in a latent neural ODE setting. However, we do believe our methodology would work in combination with other methods for handling that type of data, and we consider that an obvious avenue for future studies. We thank the reviewer for pointing out the necessity for trajectory observations as a limitation of our work at its current state, and will include this in the limitations section of the revised paper.
>
> [1] : Turan, Evren Mert, and Johannes Jäschke. "Multiple shooting for training neural differential equations on time series." IEEE Control Systems Letters 6 (2021): 1897-1902.

---

> > ### Comment · Reviewer_HMrx · 2023-08-15
> > **Reply**
> >
> > Thank you for the reply and for for clarifying a few points. As a result I increased my score by 1.

---

### Official Review · Reviewer_j2qw · 2023-07-06

**Soundness:** 3 good
**Presentation:** 3 good
**Contribution:** 2 fair
**Rating:** 6
**Confidence:** 2

**Summary:**

The authors present a novel method, Mean Inverse Integrator (MII), used to aggregate data generated through numerical integration of the vector field characterizing Hamiltonian systems. In particular, the objective is to improve the training of Hamiltonian Neural Networks (HNNs) when the data used is noisy.
The authors train HNNs on two tasks, using data generated by different methods of numerical integration.


**Strengths:**

- The paper is well written and complete. My background on the topic is limited, and so I was happy that Sections 3 and 4 were included.
- The theoretical analysis is thorough and convincing.


**Weaknesses:**

- I am not sure how much the paper fits with the themes of NeurIPS, but it should still be of interest to some of the audience.


**Questions:**

N/A

---

> ### Author Rebuttal · Authors · 2023-08-09
>
>
> We thank the reviewer for the encouraging feedback, and are happy to argue for the relevance of our paper to NeurIPS. The paper builds largely on the ideas of Hamiltonian neural networks, first presented in [1], and is related to several NeurIPS papers where numerical analysis and deep learning is combined in problems where geometry is important, such as [2,3,4,5]. Additionally, the workshop "The Symbiosis of Deep Learning and Differential Equations" has taken place in two volumes (2021, 2022) [6,7].
>
> [1] : Greydanus, Samuel, Misko Dzamba, and Jason Yosinski. "Hamiltonian neural networks." Advances in neural information processing systems 32 (2019).
>
> [2] : Finzi, Marc, Ke Alexander Wang, and Andrew G. Wilson. "Simplifying Hamiltonian and Lagrangian neural networks via explicit constraints." Advances in neural information processing systems 33 (2020): 13880-13889.
>
> [3] : Zhong, Yaofeng Desmond, Biswadip Dey, and Amit Chakraborty. "Extending Lagrangian and Hamiltonian neural networks with differentiable contact models." Advances in Neural Information Processing Systems 34 (2021): 21910-21922.
>
> [4] : Matsubara, Takashi, Ai Ishikawa, and Takaharu Yaguchi. "Deep energy-based modeling of discrete-time physics." Advances in Neural Information Processing Systems 33 (2020): 13100-13111.
>
> [5] : Chen, Yuhan, Takashi Matsubara, and Takaharu Yaguchi. "Neural symplectic form: Learning Hamiltonian equations on general coordinate systems." Advances in Neural Information Processing Systems 34 (2021): 16659-16670
>
> [6] : https://neurips.cc/virtual/2021/workshop/21880
>
> [7] : https://nips.cc/virtual/2022/workshop/49987

---

### Official Review · Reviewer_1rk9 · 2023-07-06

**Soundness:** 3 good
**Presentation:** 2 fair
**Contribution:** 2 fair
**Rating:** 4
**Confidence:** 4

**Summary:**

The presented work considers a novel class of integrators that are used to train Hamiltonian Neural Networks (HNNs). This class is called mean inverse integrator and it averages the trajectories from mono implicit RK methods (MIRK) to obtain higher accuracy. The authors provide theoretical results on how MIRK convergence suffers from noisy data. Also, experimental results on test dynamical systems illustrate the performance of the proposed approach.

**Strengths:**

The paper presents a comprehensive introduction to the different types of integrators and smoothly introduces the new one. The novel integrator is investigated w.r.t. the relations with classical RK methods (Theorem 4.4 and Proposition 4.3). Also, theoretical results on the robustness w.r.t. noise in the data are presented.

**Weaknesses:**

1) the experimental results show that the proposed approach is not always better than the alternative methods in terms of accuracy, but always slower in terms of runtime. So the use cases for the MII-related approaches should be stated more strictly.
2) the manuscript is mostly about integrators theory rather than learning dynamical systems from data. Revision of the structure and shift of the focus from integrators to its application in learning HNN can be very helpful
3) the connection of the proposed integrator and its usage in forward or in backward passes is ignored. Thus, it is unclear from the text how this new integrator should be incorporated into the existing pipeline of training HNN
4) experiments are performed only for systems, where the states are small dimensional. The scalability and robustness w.r.t the large dimension of states are unclear
5) since no batching is used for training, it indicates that a large amount of data are not tested yet and all available data are fitted in the GPU memory. It will be interesting to study how the stochasticity of the data and corresponding gradient estimates affect the convergence of the training curves.

**Questions:**

1) how do different integrators affect the way to perform backward passes? Please provide a detailed explanation and complexity analysis
2) even for 4d states the proposed approach (MII MIRK4) shows the worst runtime compared to the alternatives (Fig. 6). So, how do the MII-related integrators scale w.r.t. the state dimension? How are they applicable to learning dynamical systems with high-dimensional states?
3) all experimental results are shown for fixed noise level $\sigma= 0.05$, but authors highlight that the proposed approach is robust to noisy data. So please show how the test accuracy depends on the noise level in training data for different integrators
4) please provide analytical forms of the considered test dynamical systems for the reader's convenience

**Limitations:**

The authors explicitly state the limitations of the proposed method.

---

> ### Author Rebuttal · Authors · 2023-08-09
>
>
> We are grateful for the detailed review and suggestions for how to improve this work.
>
> ### Backward passes (Q1)
>
> Since forward-mode automatic differentiation outperforms the adjoint sensitivity method for computing derivatives of ODE solutions for smaller-sized systems ($n<100$, and we have $n=4$), this is the method chosen in this work. See for instance the discussion in [1]. Hence, the implementation is rather straightforward in the sense that solvers are implemented with PyTorch and derivatives are obtained with the standard backpropagation.
>
> It should be emphasized that the inverse injection allows the otherwise implicit MIRK methods to be implemented as explicit methods, which saves a significant amount of computations when using forward-mode AD (otherwise we would have to backpropagate through a non-linear solver or exploit the implicit function theorem). Through the repeated approximations using the inverse injection (see Appendix D), we end up with MII as a linear combination of MIRK steps. Hence, in theory the additional computational cost of MII should not be much greater than that of using MIRK directly: in the forward pass we have two matrix multiplications additionally and by the linearity of the differential operator, this should not cause a large increase in computational cost regarding the backward pass. However, as pointed out by the reviewer, MII is significantly slower than MIRK in practice. Understanding why is of high interest and we are currently investigating this in detail. A discussion on this will be included in the final version of the paper.
>
> Comparing Figure 4 and 2 in the attached PDF it is interesting to note that MII is significantly faster than RK$4$ + ISO when Adam is used as optimization method. We prefer to use L-BFGS as this yields lower error overall. However, further investigations of the difference in computational cost between Adam and L-BFGS regarding the MII will hopefully provide insight into how to improve the efficiency of the current implementation.
>
> [1] : https://www.juliabloggers.com/direct-automatic-differentiation-of-differential-equation-solvers-vs-analytical-adjoints-which-is-better/
>
> ### Scalability (Q2)
>
> We have performed some initial numerical experiments of the $3$-body problem and of the spatially discretized KdV PDE and the scalability of MII is not excellent. As mentioned above, we believe it should be possible to work on optimizing the implementation to improve this. One idea is to use a sliding-window approach where only $N$ points (and not all possible as is currently done) in positive and negative time (to the "left" and "right") are used to compute the average approximation.
>
> ### Noise level (Q3)
>
> - (Q3) The amount of noise ($\sigma$) should be considered in comparison to the average distance between two sequential points in the (unperturbed) training data:
>
> $$
> \begin{equation}
> \frac{1}{N}\sum_{n=0}^{N-1} \\|y(t_n) - y(t_{n+1}) \\|
> \end{equation}
> $$
> For the two systems and the three different step sizes $h$, this is given (computed empirically for HH and DP) by:
>
>
> *Average distance in training data*
>
> |**System**             |$h=0.4$   |$h=0.2$   |$h=0.1$   |
> |---|---|---|---|
> |Double pendulum    |$0.278$   |$0.141$   |$0.071$   |
> |Hénon-Heiles       |$0.197$   |$0.099$   |$0.050$   |
>
> This tells us that the noise level of $\sigma = 0.05$ goes from a relative magnitude of approximately $25\%$ to $100\%$ in comparison to the average distance between sequential points, meaning that the noise level is fairly high. We will clarify this in the revised version. Moreover, we will include additional experiments with noise $\sigma \in \\{0.025,0.05,0.075\\}$ to get a better understanding of the robustness of the methods. Preliminary results are found in Figure 2 in the attached PDF. Due to the limited time and having to restrict to one page, the experiments were only run for the double pendulum problem and not re-run multiple times to compute the mean and standard deviation of the error. However, the results for $\sigma = 0.05$ is consistent with Figure 6 in the paper. We observe here that MII performs on-par with RK$4$ + ISO, and significantly better for the smallest level of noise $\sigma = 0.025$.
>
>
> ### Analytical forms of vector fields (Q4)
>
> Currently this is found in Appendix A, but will be moved to section 6 for the final version which allows one page more.
>
>
> ### Batching in training
>
> We have provided experimental results using Adam and batch size of $B = 256$ in Figure 4 in the attached PDF. These may improve / differ if more time is allowed for hyperparameter optimization; however, it is interesting to note that MII is significantly faster than RK$4$ + ISO when using Adam. The error is somewhat higher using Adam than using L-BFGS (see Figure 2 with $\sigma = 0.05$ in the attached PDF).

---

> > ### Comment · Reviewer_1rk9 · 2023-08-20
> >
> > Dear authors,
> >
> > Thanks for the detailed response! I have the following comments
> > 1) The incorporation of the proposed integrator in large-scale systems is an important part of experimental evaluation. In such a setup the naive implementation of the backward pass is not enough.
> > 2) Scalability is also crucial for practical applications, so I find the current status of the work is preliminary.
> >
> > To sum up, I have decided to keep my evaluation.

---

### Official Review · Reviewer_J6Zi · 2023-07-07

**Soundness:** 3 good
**Presentation:** 3 good
**Contribution:** 2 fair
**Rating:** 5
**Confidence:** 2

**Summary:**

This paper introduces a novel method aimed at learning the vector field of a dynamical system. The proposed approach is called the mean inverse integrator, which utilizes a neural network (e.g., SRNN) to accurately estimate the integrator in the presence of noisy data. The authors provide theoretical insights into the sensitivity of both the one-step target function and the mean inverse integrator to data noise. Additionally, the paper presents empirical evaluations by comparing the method to five different types of integrators.

**Strengths:**

- The paper effectively conveys its ideas and arguments with clarity.

- The research addresses an important question and presents a new approach to handle noise when learning data dynamics.

- Theoretical analysis shows how the proposed method and the baseline approach respond to noise, contributing to a better understanding of their performance.

- Empirical results demonstrate the effectiveness of the proposed method, showing significant advantages over the baseline approaches.

**Weaknesses:**

- The paper lacks clarity on why the mean inverse integrator yields more accurate estimates compared to the one-step baseline.

- Scalability to high-dimensional systems is not explored, as current experiments primarily focus on trivial test cases. It would be beneficial to investigate potential challenges arising from increased computational complexity, instability of estimation in the presence of chaotic dynamics, and provide examples demonstrating the method's efficacy.

**Questions:**

Q1: The paper would benefit from running experiments on complex, high-dimensional data to showcase the method's capabilities.

Q2: It remains unclear whether the proposed method can generalize to more general dynamical systems beyond Hamiltonian systems.

---

> ### Author Rebuttal · Authors · 2023-08-09
>
> We thank the reviewer for the feedback and would like to provide the following response.
>
> - The idea of the theoretical analysis behind Theorem 5.2 is precisely to provide an understanding of why the mean inverse integrator provides more accurate estimates, namely because the averaging over multiple approximations allows noise to be canceled out.
> - (Q1) The reviewer has a fair point in that the systems used for numerical experiments are low dimensional. However, both Hénon-Heiles and the double pendulum problem do exhibit chaotic dynamics and the FPUT problem is highly oscillatory with fast oscillations along some dimensions and slow oscillations among others.
> - (Q1) For tackling higher-dimensional systems (such as spatially discretized PDEs, the N-body problem or a coupled spring system) the methodology presented would have to be extended by considering other neural network architectures (i.e. CNNs for PDEs) or considering the problems along other coordinate systems, such as studied by Finzi et al. [1]. We consider this a highly relevant future work, and we have indeed performed experiments on the discretized KdV equation and the 3-body problem, but decided that it is outside the scope of this paper. This is partly because we wish for our results to be compared to the recent literature on Hamiltonian neural networks, where the focus has been on lower-dimensional systems, and partly because considering higher-dimensional systems adds complexity and further considerations about the neural networks used that may cloud the presentation and analysis of how our methods perform compared to benchmarks.
> - (Q2) The proposed method does apply to learning vector fields in any form. We chose to focus on Hamiltonian systems in the paper so our method could be linked to and compared to the vast recent research on Hamiltonian neural networks, and to demonstrate that non-symplectic integrators may work very well also here. To demonstrate the utility of our method for general systems, we have included numerical experiments for the Lotka-Volterra system in Figure 1 in the attached PDF. Here, we see that MII is significantly better than ISO + RK$4$ and the MIRK methods are generally superior. Since the Lotka-Volterra system is not a canonical Hamiltonian system, the vector field is learned directly with a neural network $f_{\theta} : \mathbb{R}^2 \rightarrow \mathbb{R}^2$.
>
> The Lotka-Volterra problem is given by the following ODE:
> $$
> \dot x_1 = x_1 - x_1x_2,
> $$
> $$
> \dot x_2 = x_1x_2 - x_2.
> $$
>
> [1] : Finzi, Marc, Ke Alexander Wang, and Andrew G. Wilson. "Simplifying Hamiltonian and Lagrangian neural networks via explicit constraints." Advances in neural information processing systems 33 (2020): 13880-13889.

---

> > ### Comment · Reviewer_J6Zi · 2023-08-20
> >
> > I acknowledge the reading of the authors’ replies and am aware of the discussion related to the noise scales. I decide to keep my score.
> > Thank you!

---

### Author Rebuttal · Authors · 2023-08-09

Here we attach a PDF with four additional numerical experiments, responding to specific questions posed by the reviewers. The figures are referenced in the rebuttals below.

---

### Decision · Program_Chairs · 2023-09-21

**Decision:**

Reject

**Comment:**

Although most of the reviewers find an interesting and challenging point in this paper and have positive impressions of this paper, they also have concerns about the significance of the content of this paper. Also, the connection or usefulness of this paper to ML problems is not necessarily clearly described. In the discussion phase, the authors provide useful feedback on some of the concerns raised by the reviewers. However, it seems that their concerns have not been adequently addressed. Overall, I recommend rejection of this paper for the current submission.